# Reagent-Free Immobilization of Industrial Lipases to Develop Lipolytic Membranes with Self-Cleaning Surfaces

**DOI:** 10.3390/membranes12060599

**Published:** 2022-06-09

**Authors:** Martin Schmidt, Andrea Prager, Nadja Schönherr, Roger Gläser, Agnes Schulze

**Affiliations:** 1Leibniz Institute of Surface Engineering (IOM), Permoserstr. 15, 04318 Leipzig, Germany; martin.schmidt@iom-leipzig.de (M.S.); andrea.prager@iom-leipzig.de (A.P.); nadja.schoenherr@iom-leipzig.de (N.S.); 2Institute of Chemical Technology, Leipzig University, Linnéstraße 3, 04103 Leipzig, Germany; roger.glaeser@uni-leipzig.de

**Keywords:** enzyme membrane reactor, lipase, fouling, self-cleaning surface, electron beam, response surface methodology

## Abstract

Biocatalytic membrane reactors combine the highly efficient biotransformation capability of enzymes with the selective filtration performance of membrane filters. Common strategies to immobilize enzymes on polymeric membranes are based on chemical coupling reactions. Still, they are associated with drawbacks such as long reaction times, high costs, and the use of potentially toxic or hazardous reagents. In this study, a reagent-free immobilization method based on electron beam irradiation was investigated, which allows much faster, cleaner, and cheaper fabrication of enzyme membrane reactors. Two industrial lipase enzymes were coupled onto a polyvinylidene fluoride (PVDF) flat sheet membrane to create self-cleaning surfaces. The response surface methodology (RSM) in the design-of-experiments approach was applied to investigate the effects of three numerical factors on enzyme activity, yielding a maximum activity of 823 ± 118 U m^−2^ (enzyme concentration: 8.4 g L^−1^, impregnation time: 5 min, irradiation dose: 80 kGy). The lipolytic membranes were used in fouling tests with olive oil (1 g L^−1^ in 2 mM sodium dodecyl sulfate), resulting in 100% regeneration of filtration performance after 3 h of self-cleaning in an aqueous buffer (pH 8, 37 °C). Reusability with three consecutive cycles demonstrates regeneration of 95%. Comprehensive membrane characterization was performed by determining enzyme kinetic parameters, permeance monitoring, X-ray photoelectron spectroscopy, FTIR spectroscopy, scanning electron microscopy, and zeta potential, as well as water contact angle measurements.

## 1. Introduction

Biocatalysis is becoming increasingly important as a sustainable approach in chemistry and biotechnology. The global market for industrial enzymes is estimated to be worth US $4.5 billion, with projections of $7 billion by 2021 [1]. Enzymes are known to be catalytically highly active and selective, but unlike conventional chemical catalysis, they usually operate under mild conditions. Free enzymes in solution tend to aggregate, which reduces their catalytic activity, selectivity, stability, and reusability. Immobilization on a support material can be advantageous for a large range of applications [2,3,4].

The combination of (polymer-based) membrane filters and enzymes is known as enzyme membrane reactor (EMR) or biocatalytic membrane reactor (BMR) [5]. The principle of BMRs is based on the separation of the catalyst and the biotransformation products or substrates by a selective material. Here, the membrane can act as a separation unit, thus protecting the permeate stream from contamination with the catalyst, or it can act as a support layer for immobilization of the enzyme molecules [6,7,8]. In recent years, many applications emerged, e.g., for degradation of biomacromolecules [9,10], hydrolysis of pollutants or pharmaceuticals [11,12], but also for the synthesis of compounds [13,14].

In particular, hydrolytic enzymes (hydrolases) such as proteases for the degradation of proteins, lipases for the degradation of oils and fats, as well as glycosidases for complex sugars are suitable for the production of bioactive anti-fouling membranes [15,16,17,18]. Enzymes as biocatalysts are able to cleave bonds within bio(macro)molecules in the presence of water, producing much smaller fragments. Hence, a self-cleaning capability can be introduced to provide mild cleaning, reduce the usage of chemical cleaning agents, and extend the life of the membrane material. Many different ways of immobilization have been investigated, e.g., chemical-based coupling reactions [19]. However, this is associated with serious disadvantages, such as high costs due to expensive coupling chemicals, ineffective production due to very long reaction times, or the risk of contamination with toxic or hazardous reagents.

In recent years, an advanced radiation-initiated grafting technology based on electron beam (EB) irradiation has been developed [20]. This method was used to immobilize small organic molecules [21], large photoactive dyes [22], synthetic polymers [23], peptides and proteins [24,25], and even enzymes [26,27] on polymer materials such as polyvinylidene fluoride (PVDF), or polyethersulfone (PES), respectively. The membrane is first impregnated with an aqueous solution of a grafting compound and then irradiated with high-energy radiation such as electron beams. Activation of the polymer substrate and solutes occurs simultaneously, resulting in rapid radical-based reactions that modify the membrane surface. Advantages of this approach are clean processing requiring only an aqueous solution, fast reaction rates enabling continuous mode of operation, and a wide range of polymer-based substrates as well as graft compounds. A drawback might be the stochastic, i.e., non-directed immobilization of the modifiers.

In this study, industrial lipases were attached to PVDF flat sheet microfiltration (MF) membranes using electron beam irradiation. By applying the response surface methodology (RSM) in a design-of-experiments (DoE) approach, the effects of three numerical factors on the immobilized enzyme activity have been investigated. The enzyme-loaded membranes exhibited lipolytic behavior, resulting in a bioinspired BMR for lipid degradation. The self-cleaning capability was tested with fouling experiments using olive oil, and the reusability in three consecutive cycles was demonstrated.

## 2. Materials and Methods

### 2.1. Materials

All samples were based on a commercially available polyvinylidene fluoride flat sheet microfiltration membrane (PVDF; ROTI^®^, 0.45 μm), purchased from Carl Roth (Karlsruhe, Germany). Two industrial lipases, FE-01 and EL-01, were obtained from ASA Spezialenzyme GmbH (Wolfenbüttel, Germany). The enzymes were provided in an unspecified buffer solution containing additives such as glycerol (EL-01: 20–30%; FE-01: 45–49.5%). Please note that both are the same lipase enzyme, with EL-01 being characterized by more purification steps and fewer stabilizing additives. The origin was stated to be *Thermomyces lanuginosus* (fungus), recombinant from *Aspergillus oryzae* (fungus). Triton X-100 detergent and absolute ethanol (EtOH) were purchased from Merck (Darmstadt, Germany). Enzyme activity assays were performed with 4-nitrophenyl laurate (pNP-C12) in BIS-TRIS propane buffer (BTP; 50 mM BTP, and each 1 mM CaCl_2_, MgCl_2_, NaCl, and KCl, respectively), obtained from Sigma Aldrich (St. Louis, MO, USA). Enzyme quantification was performed using a commercial assay (Pierce^™^ BCA Protein Assay Kit) from Thermo Scientific (Rockford, IL, USA), employing bovine serum albumin (BSA) as a reference protein. Fouling tests were carried out by using extra virgin olive oil (“Gentile”, Bertolli, Italy), acquired from a local grocery store. Deionized water in Millipore^®^ quality was used; all chemicals were employed without further purification.

### 2.2. Enzyme Immobilization

In general, the procedure of this electron beam-induced immobilization is based on results from recent work and was carried out as described therein [25]. Briefly, a hydrophobic PVDF sample (Ø = 33 mm) was pre-wetted using EtOH for about 0.5 min, followed by an exchange with water for at least 4 × 5 min. The pre-wetted membrane was impregnated in 4 mL of a freshly prepared enzyme solution at room temperature and without shaking for a specified time (*cf.*
Section 2.4). Subsequently, the sample was removed and irradiated in a wet but slightly drained state utilizing a self-built electron accelerator. The irradiation was performed in the N_2_ atmosphere (O_2_ < 15 ppm) at 2.5 m min^−1^ conveyor speed, respectively. A specified irradiation dose was applied by adjusting the beam current (*cf.*
Section 2.4). Finally, the irradiated membrane was washed with 5 mL Triton X-100 (ω = 5%; 10 min) and 4 × 10 min with water in excess by shaking at 450 rpm.

### 2.3. Enzyme Activity

The enzyme activity was investigated by performing a continuous photometric assay employing pNP-C12 as substrate. The assay is based on the cleavage of the substrate via hydrolysis of the ester bond leading to the release of 4-nitrophenol (pNP), which is a yellow dye at pH 8. The reaction was monitored continuously by using a microplate reader (Infinite M200, Tecan, Austria). Briefly, an enzyme-loaded membrane sample (Ø = 33 mm, with a central hole of Ø = 10 mm) was inserted into a 3D-printed scaffold and placed in a 6-well microplate (Appendix A). Subsequently, 4.0 mL of the substrate solution (200 µM pNP-C12 with φ = 15% DMSO in BTP buffer, pH 8) was added. Immediately, the recording was started, and the absorbance at 405 nm was measured in intervals between 15 and 60 s. The obtained progress curves were analyzed using the DynaFit 4.09 software (Petr Kuzmic (BioKin, Ltd.), Watertown, MA 02472, US) package [28,29], employing a self-made script (Appendix A). The DynaFit approach allows the accurate and efficient determination of important enzyme kinetic parameters by fitting the integrated closed-form Michaelis-Menten rate equation, Equation (1) [30,31,32]:(1)$$\left[P\right]=\left[S_{0}\right]-K_{m}\cdot W\left\{\frac{S_{0}}{K_{m}}\cdot \exp\left(\frac{S_{0}-V_{max}\cdot t}{K_{m}}\right)\right\}$$
where [*P*] is the determined concentration of the released product, pNP; [*S*_0_] is the initial concentration of the substrate, pNP-C12; *K_m_* is the apparent Michaelis constant, *W* is the Lambert W function; *V_max_* is the apparent limiting reaction rate, and *t* is the time. Please note that an additional correction term was used in this study to account for non-specific effects (*cf.* Appendix A). For converting the observed absorbance into a concentration, a calibration curve using pNP was created (0, 25, 50, 100, 150, and 200 µM).

### 2.4. RSM Design

In order to investigate the immobilization process, a response surface methodology (RSM) within statistics software Design-Expert 13 (Stat-Ease, Minneapolis, MN, USA) was applied [33]. Initially, a central composite design with four central points was created, resulting in a total of 36 experimental runs, i.e., 18 runs per enzyme. After the first evaluation, a significant lack-of-fit was detected, and the design was augmented to a full cubic I-optimal design. Hence, 12 runs were added, resulting in 48 total experiments. The investigated factors consisted of 3 numerical parameters: mass concentration of the enzyme solution, impregnation time, and irradiation dose. Additionally, both enzymes were considered as one categoric factor. The factor levels are provided in Table 1. The examined response was the immobilized enzyme activity in terms of the apparent limiting reaction rate, *V_max_*, determined via progress-curve analysis (*cf.*
Section 2.3). A fraction of the design space score of 84% (S/N = 2) or 100% (S/N = 3), respectively, proved the high quality of the design. After performing all measurements, the RSM design was analyzed via analysis of variance (ANOVA) and reduced by selecting only terms with a *p*-value ≤ 0.05. Finally, the reduced model was utilized to optimize the enzyme activity by employing the desirability function within Design-Expert [34]. In this way, optimal conditions for enzyme immobilization can be determined based on the empirical data obtained in the design-of-experiments approach. Optimized settings were applied to confirm the model by running five replicate runs for the enzyme-loaded membrane.

### 2.5. Fouling and Self-Cleaning

All tests were carried out according to preliminary works [17,18,35]. In general, the aim was to generate severe fouling, i.e., a highly clogged membrane. Subsequently, the self-cleaning capability introduced by the immobilized hydrolases was monitored by determining the regeneration of filtration performance at regular intervals. Thus, the dead-end filtration model was chosen, as well as a relatively high concentrated olive oil feed (1 g L^−1^ in 2 mM sodium dodecyl sulfate, SDS). Briefly, one fouling cycle consisted of alternating 4 × 500 mL filtration with olive oil and 4 × 100 mL backwashing with water, respectively. At the beginning and after the fouling cycle, the pure water permeance was measured to determine the degree of fouling. Enzyme activation was then performed by placing the fouled samples in 10 mL of PBS buffer (pH 8) in an oven (37 °C). After every 0.5 h, the water permeance was measured to estimate the degree of regeneration. The filtration performance was monitored for each step by measuring the permeation time for the first 100 mL and calculating the permeance according to Equation (2). Experiments were carried out in triplicate using membrane samples with Ø = 44 mm (effective area considering the sealing ring: 11.9 cm²) and a stirred filtration cell (200 rpm; Amicon, Merck Millipore, Billerica, MA, USA).
(2)$$P=\frac{V}{A\cdot t\cdot p}$$
where *P* is the permeance, *V* is the filtered volume (first 100 mL), *A* is the effective membrane area (11.9 cm²), *t* is the permeation time, and *p* is the applied pressure (1 bar).

### 2.6. Characterization

Comprehensive methods were carried out to characterize the untreated reference and the optimized enzyme-loaded membrane. Detailed descriptions and settings were reported elsewhere [25].

*Pure water permeance (P_w_)*: The pure filtration performance was calculated from the measured permeation time for filtering 200 mL water at a pressure of 1 bar using a stainless-steel filtration cell (16249, Sartorius Stedim Biotech, Göttingen, Germany).

*Grafting yield (GY)*: The amount of immobilized enzyme was determined using a commercial assay based on bicinchoninic acid (BCA) [36]. As the employed enzymes were provided by the manufacturer without a specified concentration, bovine serum albumin (BSA) from the assay kit was used as a reference for both the crude and the immobilized enzyme, respectively (*n* = 5).

*X-ray photoelectron spectroscopy (XPS)*: The chemical surface composition of pristine and EL01-modified PVDF membranes was investigated by XPS. Furthermore, the fouling tests were supported by XPS measurements before and after the first cycle. All measurements were performed at least in quadruplicate.

*Water contact angle (WCA)*: The surface wettability with water was determined by applying the sessile drop method and the Young–Laplace model (*n* = 5).

*Fourier-transform infrared spectroscopy (FTIR)*: Transmittance IR spectroscopy was applied to estimate the fouling degree and self-cleaning capability before and after the first cycle. All measurements were performed at least in triplicate.

*Zeta potential*: The pH-dependent surface charge was determined by measuring the streaming potential in an adjustable gap cell and calculating the zeta potential (*n* = 4).

*Scanning electron microscopy (SEM)*: The surface morphology of untreated and modified samples, as well as before and after the first fouling and self-cleaning cycle, was observed with SEM. Multiple spots were analyzed with magnifications ranging from 300- to 25,000-fold to provide a representative overview.

## 3. Results and Discussion

### 3.1. Enzyme Immobilization

Two industrial lipase enzymes were coupled on PVDF flat sheet membranes using electron beam (EB) irradiation as a reagent-free immobilization method. A design-of-experiments (DoE) approach was performed to investigate the effects of four process factors on the immobilized enzyme activity (Table 1). Briefly, the response surface methodology was employed, using statistics software Design–Expert 13 to generate an I-optimal design that is characterized by minimizing the average variance of prediction [33,37]. Each experiment consisted of a combination of factor levels within predefined limits, resulting in 48 enzyme-loaded membrane samples. Subsequently, a kinetics assay using 4-nitrophenyl laurate (pNP-C12) was performed [38]. The release of the dye 4-nitrophenol (pNP) by hydrolysis was continuously recorded by measuring the absorbance at 405 nm. The software package DynaFit (Petr Kuzmic (BioKin, Ltd.), Watertown, MA 02472, US) [28,29] was applied to estimate important kinetic parameters from the progress curves according to Equation (1) and Appendix A. Finally, the I-optimal design was used to optimize the limiting reaction rate, *V_max_*. In this way, the optimal condition for the enzyme immobilization was obtained (cf. Section 2.4). The complete design is listed in Appendix A.

In this study, two industrial lipases were selected as promising sources for highly potent yet inexpensive enzymes. Most likely, the purchased products consisted of mixtures of different carboxylester hydrolases. A distinction can be made between lipolytic esterases (“lipases”) and nonlipolytic esterases [39]. Their mode of action, i.e., hydrolysis of an ester bond, is their common feature, which is why they are classified as EC 3 according to the Enzyme Commission of the International Union of Biochemistry and Molecular Biology [40]. However, structural differences within protein domains, as well as differences in substrate specificity, provide a distinction [41]. Nevertheless, both purchased enzymes were essentially from the same origin, with the enzyme EL-01 being more concentrated and further purified. Boxplot analysis revealed that the average *V_max_* of both immobilized enzymes was the same, with EL-01 having higher maximum values (Appendix A). The latter might be due to fewer constituents in the buffer acting as potential radical scavengers during irradiation.

The model was analyzed via analysis of variance (ANOVA) and reduced with a criterion of *p*-value ≤ 0.05 to retain only the significant factors. The final model consisted of 17 terms, including all main effects, two-factor interactions, and quadratic and cubic terms, respectively (Table 2). The model F-value of 41.24 implies the model is highly significant. An adjusted correlation coefficient of $R_{adj.}^{2}=0.9370$ indicated a strong correlation, and a prediction coefficient of $R_{pred.}^{2}=0.8720$ proved a very good prediction capability. The lack-of-fit was not significant, i.e., there was no need to use higher-order models. The data had to be transformed with the natural logarithm, *ln*, to satisfy Gauss–Markov criteria (Appendix A).

Since the average reaction rates were similar, but the maximum rates were slightly higher for the enzyme EL-01, the discussion here is limited to this lipase. The response surface of the EB-based grafting process revealed a strong dependence on all main factors, but especially on irradiation dose and enzyme concentration (Figure 1). Most importantly, the activity decreases rapidly with a higher EB irradiation dose. An optimum is observed below a dose of 100 kGy. Interestingly, previous studies with small organic molecules [20,42], larger compounds [22,23], as well as with the protein BSA [25] have shown that EB doses of 100–150 kGy resulted in higher graft yields. Even the EB-based coupling of the protease trypsin [26] and the oxidoreductase laccase [43] exhibited higher activities at 150 kGy. In contrast, the present study confirmed that there was hardly any lipase activity above 150 kGy.

This difference can be explained by the composition of the impregnation solutions. While in other works [20,23,25,43], the purified compounds were dissolved in pure water. In this study, technical enzymes were used in buffer solution as provided by the manufacturer. In particular, glycerol was present in very high concentrations (FE-01: >45%; EL-01: 20–30%). Thus, it is most likely that the buffer compounds interfere with the coupling process by scavenging the water radiolysis products or by undergoing radical recombinations. In addition, lipases and esterases are known to be more sensitive to ionizing radiation than other enzymes [44]. Lipases are characterized by a hydrophobic region near the active site (“lid”), which acts as an interface between the aqueous phase and the lipid phase [41]. Radiolysis products could attach to the protein structure, leading to hydrophilization [45].

Recently, a mechanism for this radiation-induced graft immobilization (RIGI) approach was proposed [42]. Briefly, EB irradiation leads to activation of the PVDF polymer by the formation of reactive radical sites or, to a lesser extent, alkene groups [46,47]. Furthermore, water radiolysis products such as OH radicals (~2.6 per 100 eV) or solvated electrons (2.7 per 100 eV) are formed [48]. These highly reactive short-lived species can perform H-abstractions, or additions, respectively, resulting in activation of the solutes (graft compounds) through radical chain reactions. Typically, at low solute concentrations (<1%), their radiation chemistry is of minor importance because the solvent takes up most of the absorbed energy. Thus, the chemical reactions are indirectly driven by radiolytic solvent species. However, in the case of very high concentrated compounds such as glycerol in this study, additional reaction pathways due to “direct actions” have to be considered [49]. Hydroxyl radicals are reactive towards alcohols, typically performing α-carbon H-abstractions. Hence, it has been suggested for glycerol that direct actions from γ-rays lead to the formation of smaller fragments such as formaldehyde, acetaldehyde, methanol, acetone, and acetol, respectively [50]. These compounds are able to compete with substrates for the active site of the enzyme, resulting in competitive inhibition [51].

Regarding the enzyme concentration of the impregnation solution, the evaluation showed that the reaction rate increases with higher concentration, and an optimum is reached at about 8 g L^−1^ (≈0.8 wt.−%). Interestingly, it appears that macromolecules may benefit from high concentrations, while smaller molecules perform best at about 0.1 wt.−% [20,42]. This is consistent with previous studies on BSA, which resulted in steadily increasing graft yields at up to 18 g L^−1^, presumably attributed to crosslinking events resulting from the formation of multiple reactive sites [25,52,53]. Macromolecules provide a larger surface for the attack by water radiolysis products, which would facilitate crosslinking, while small molecules would be more likely to be inactivated by dimerization due to their high mobility. However, in the case of EL-01 immobilization, the observed activity decreased above 8 g L^−1^. First, crosslinking most likely affects the tertiary structure and thus activity. Second, as the enzyme concentration increases, the glycerol content also rises, which promotes additional competitive reactions that may lead to less grafting or inhibition, respectively.

Finally, a positive effect of impregnation time was found. Notably, the significance of the higher-order model terms is greater than that of the main effect (*cf.*
Table 2, *p*-values of B², B³ *vs*. B). Short impregnation of 1 min resulted in hardly any activity. Interestingly, studies with BSA showed appreciable amounts were immobilized after only 6 s of impregnation [25]. In the case of industrial lipases, the buffer additives are probably the cause of lower grafting yields, delaying exposure to the PVDF polymer chains. Nevertheless, the maximum hydrolysis rate was found at impregnation times of about 5 min, which is well below the typical chemical coupling reaction time of hours [54].

To determine the maximum reaction rate for the immobilized EL-01, optimization via desirability was carried out using Design-Expert 13 [34]. The prediction suggests an enzyme mass concentration of *β_opt_* = 8.4 g L^−1^, impregnation time of *t_opt_* = 5 min, and a dose of D_opt_ = 80 kGy (Figure 1c). The model was confirmed by repeating five experimental runs and calculating the kinetic parameters *V_max_*, *K_m_*, and *V/K* using the DynaFit software (Table 3). The obtained *V_max_* value of 2.7 ± 0.4 µM s^−1^ was within the 95% prediction interval (PI; 1.45–3.39 µM s^−1^; predicted mean: 2.34 µM s^−1^). One lipase unit (U) is defined as the quantity of enzyme that will release 1 μmol of pNP per minute under the conditions of the test (pH 8.0, 25 °C) [55]. The specific enzyme activity was determined to be 823 ± 118 U m^−2^ (about 11.4 U g^−1^). The progress curve of the optimized PVDF-*g*-EL membranes is given in Figure 1d showing a fast conversion of the pNP-C12 within 150 s. In contrast, the untreated PVDF-Ref samples showed no hydrolytic activity.

**Figure 1 membranes-12-00599-f001:**
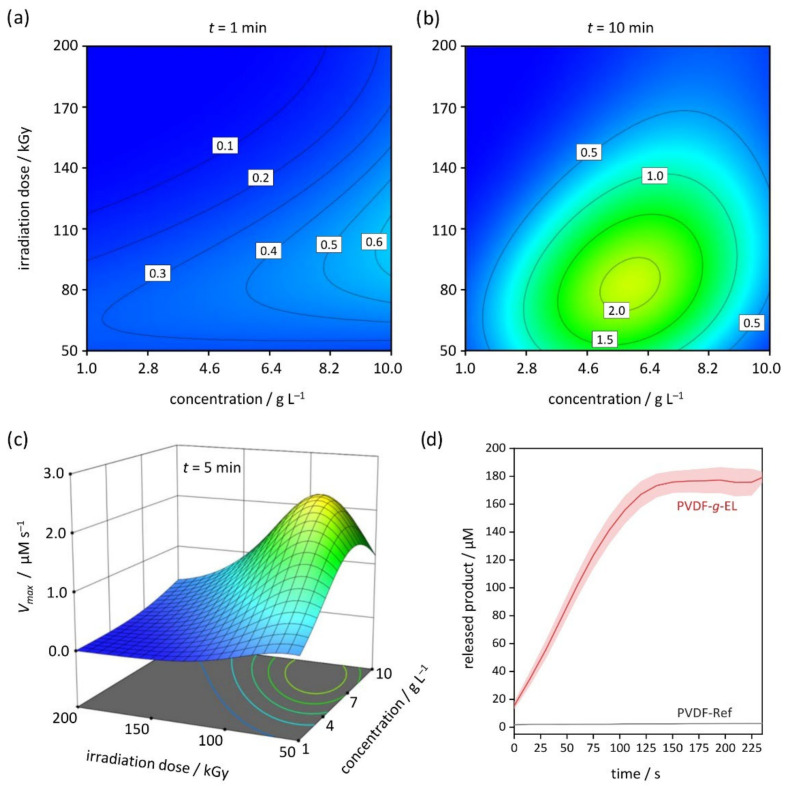
Evaluation of *V_max_* (I-optimal RSM design). Contour plots for enzyme concentration and irradiation dose at (**a**) 1 min; (**b**) 10 min; and (**c**) 3D surface plot at 5 min; as well as (**d**) progress curves of optimized PVDF-*g*-EL and pristine PVDF-Ref (*n* = 5).

### 3.2. Characterization

The efficiency of immobilization was examined by characterizing optimized PVDF-*g*-EL samples. XPS measurements confirmed the presence of enzyme as the N content increased from 0% to 0.58% due to amino acids (Table 4). The decrease in the elemental ratio F/C from 62.89% to 52.47% indicated the formation of a layer covering the PVDF polymer chains. Almost no differences were found between the top and bottom sites of the membrane sample. However, the total graft yield obtained by XPS was relatively low. In addition, the BCA assay, which determines the total amount of protein, also showed relatively low amounts of 31.2 ± 11.7 mg m^−2^. It should be noted that the values were determined against BSA as a reference substance, which means that deviations from the real protein content are to be expected because of different amino acid compositions [56]. Most likely, the aspects addressed in Section 3.1 are responsible for the low graft yield.

Significant changes in surface properties were detected via zeta potential and water contact angle measurements. The zeta potential was shifted to higher values, and below pH 5 it is even positive (Figure 2a). Furthermore, the isoelectric point (IEP) shifted from 3.8 to 5.0. Thus, the surface charge became more positive overall due to enzyme immobilization. In contrast, the untreated reference has a negative surface charge over a broader pH range, which is caused by the adsorption of OH anions [57]. The water contact angle of PVDF-*g*-EL samples decreased from about 143° to 113° (top) or 122° (bottom), respectively, demonstrating the EB penetration capability (Figure 2b). Thus, the samples became less hydrophobic, which usually has a positive effect on long-term performance [58,59]. However, the reduction was not very strong, which is consistent with the amount of attached enzyme. The morphology was not altered during the treatment, and no blockages or defects could be observed in SEM images. Furthermore, the pure water permeance *P_w_* did not change: PVDF-Ref, 17,713 ± 77 L m^−2^ h^−1^ bar^−1^, *vs.* PVDF-*g*-EL, 17,637 ± 143 L m^−2^ h^−1^ bar^−1^, respectively.

### 3.3. Fouling and Self-Cleaning

Self-cleaning polymeric membranes prepared by chemical-based grafting reactions of various hydrolases have been demonstrated in previous work [18,35,60]. In this study, a lipolytic hydrolase was selected to evaluate the electron beam-based BMR fabrication. Fouling tests were carried out with olive oil mixtures (1 g L^−1^ in 2 mM SDS). The objective of the experiment was to generate severe fouling and subsequently monitor the regeneration of water permeance. Thus, a dead-end filtration mode was selected, as the entire feed volume is passed through the filter, which promotes clogging. To simulate a long-term operation, backwashing with water was performed after each olive oil step to remove only slightly bound foulants. A fouling cycle consisted of filtering a total of 2 L of olive oil and 0.4 L of water as backwash. In addition, 0.1 L of water was filtered at the beginning and after the last backwash, respectively, to determine the fouling degree based on the decrease in water permeation. Finally, the self-cleaning cycle was started by enzyme activation via impregnation in an aqueous PBS buffer (pH 8, 37 °C). After every 0.5 h, the water permeance was measured to estimate the regeneration.

All tests were performed in triplicate, using the pristine PVDF membrane and the optimized PVDF-*g*-EL (Figure 3). Please note that because of a different filtration setup and stirring, the permeance values are lower than the ones reported in Section 3.2. Within one cycle of fouling, the water permeance of the PVDF-Ref membrane decreased rapidly from 11,312 ± 265 to 202 ± 11 L m^−2^ h^−1^ bar^−1^. This corresponds to a decline of 98%, clearly indicating severe fouling due to the formation of a lipid layer. In an industrial application, such a filter would have to be intensively cleaned with chemical agents such as NaOH, HCl, or citric acid [61]. In contrast, the PVDF-*g*-EL samples showed a much smaller decrease in permeance from 13,066 ± 159 to 4183 ± 642 L m^−2^ h^−1^ bar^−1^ (32% of the initial value). This trend is consistent with chemically immobilized enzymes showing a lower degree of fouling [18]. Two reasons can be considered: (a) enzymes might already be active during the filtration providing an *in-situ* cleaning; and (b) the modification leads to improved surface properties, particularly reduced hydrophobicity, which enables passive fouling reduction. The latter is supported by the reduced water contact angle (*cf.*
Section 3.2). Hydrophilization strategies are a well-known approach to passively reduce fouling in membrane filters, as they reduce the likelihood of hydrophobic interactions between foulants and the polymer material [62]. Enzyme activation by adjusting pH and temperature is a common way to increase enzyme activity, as most enzymes are highly dependent on both parameters. In the case of EL-01, the optimum activity of the crude enzyme is at pH 10.5 and 40 °C, according to the manufacturer. Hence, sufficient activity can be expected under the conditions of the test.

Subsequently, during the self-cleaning cycle, the PVDF-Ref could not regenerate its surface, resulting in final permeance of 548 ± 28 L m^−2^ h^−1^ bar^−1^, which corresponds to 5% of the initial permeability. Even after 24 h in the oven at 37 °C, the value raised to only 7%. In comparison, the PVDF-*g*-EL samples showed a complete regeneration within 3 h. SEM images visually confirmed the measurements by showing a clean surface in contrast to the clearly visible lipid layer on the fouled PVDF-Ref samples (Figure 4).

Interestingly, an opposite effect was observed within the first 0.5 h of cleaning, as the permeance of the PVDF-*g*-EL sample decreased rapidly, indicating complete clogging of the pores. Since olive oil is mainly composed of triacylglycerols (triglycerides or fats), the initial hydrolysis of an ester bond leads to the formation of a diglyceride and a free fatty acid (FFA). Subsequently, further hydrolysis leads to monoglycerides and, finally, to free glycerol, releasing more fatty acids [63]. Mono-and diglycerides are capable of producing gel-like phases that entrap liquid oil [64,65,66,67]. The initial decrease in permeance can most likely be explained by the formation of gel-like particles in the pore volume of the membrane. Nevertheless, as the reaction progressed, i.e., increasing hydrolysis of the glycerides, combined with the steady replacement of the activation buffer, the permeance regenerated at a high rate shortly thereafter. Thus, for the actual application of self-cleaning membranes, active washing by filtration is preferable to passive impregnation in order to facilitate the removal of degradation products.

Moreover, the results were supported by XPS and FTIR spectroscopy: After the first cycle, the PVDF-Ref sample showed a sharp decrease in F content from 37.2 ± 1.1% to 4.2 ± 0.6% while increasing the O content from 3.1 ± 0.4% to 12.1 ± 0.6% due to formation of a lipid layer. In contrast, the PVDF-*g*-EL sample showed a much smaller decline in F content from 32.3 ± 1.3% to 20.4 ± 1.7% and only a slight increase in O content, indicating a much lower amount of foulants (Figure 5a). A minimal increase in N, S, and Si content was observed in all samples after fouling, most likely due to components within the olive oil. In addition, similar observations were made for the top and bottom sites. Full data are listed in Appendix A. FTIR transmission spectra (Figure 5b) confirmed the presence of olive oil based on the appearance of a sharp signal at 1745 cm^−1^ (stretching vibration of C=O groups of the triglycerides) and signals at 2925 and 2855 cm^−1^ (stretching vibrations of aliphatic C-H in CH_2_ and CH_3_ groups of the fatty acid residue) [68]. Signals in the fingerprint region of 1466–960 cm^−1^ are assigned to bending and rocking vibrations of aliphatic groups (not shown) [68]. Please note that due to the low abundance of the enzyme (XPS: N = 0.58%), the typical amide bands are very weak in the FTIR spectra. In summary, all signals were much stronger in the reference sample than in the PVDF-*g*-EL sample, indicating that the lipid layer was largely removed in the latter. However, some remaining degradation products are still present, but without affecting the filtration performance.

The reusability of the enzyme-loaded PVDF samples was evaluated by performing three consecutive fouling and self-cleaning cycles (Figure 6). Since the reference membrane was already severely clogged after one cycle, only the PVDF-*g*-EL sample was used. This setup corresponds to a total filtration volume of about 7.8 L (without backwashing). Thus, the experiment is equivalent to an upscaled filtration of 6550 L m^–2^. The results show that even after 3 cycles of olive oil fouling and subsequent self-cleaning, a regeneration of the water permeance to 102 ± 2% (2nd cycle), and 95 ± 3% (3rd cycle), respectively, could be achieved. For all cycles, the self-cleaning time was set to 3 h with monitoring every 0.5 h. Hence, the prepared lipolytic membrane reactors showed an excellent self-cleaning capability with promising reusability in long-term usage. Nevertheless, an increasing fouling and a delayed regeneration are observable. More sophisticated setups are needed to estimate the performance under real conditions, e.g., by applying more frequent and extended backwashing steps. Although olive oil is a natural product containing a variety of different components, including non-lipids such as tocopherols, chlorophyll, or phenolic compounds [69], it can be considered relatively pure. In real-world applications, much greater variability of compounds can be expected, which might also inhibit enzymes [70]. It is difficult to estimate the lifetime of PVDF-*g*-EL membranes, but the enzyme activity generated in this study was obtained after washing with a solution of 5% Triton X-100 detergent. This is 10-fold higher than, e.g., the washing required to remove BSA protein from PVDF [25]. Thus, it can be assumed that the EB-treated lipase membranes have a higher resistance to delamination by detergents than an immobilization via, e.g., adsorption. Potentially, a combination of enzymes could be utilized to cope with a wide range of foulants.

## 4. Conclusions

In this study, a reagent-free immobilization of industrial lipases was performed by applying electron beam irradiation. PVDF microfiltration membranes were modified with hydrolases to generate a biocatalytic membrane reactor with lipolytic properties. A response surface methodology-based experimental design was employed to investigate the effects of three numerical parameters: enzyme concentration, impregnation time, and irradiation dose. An optimized procedure (8.4 g L^−1^, 5 min, 80 kGy) with a specific enzyme activity of 823 ± 118 U m^−2^ is presented. Notably, the short impregnation times and the fast radical-based reaction mechanism within a fraction of seconds allow upscaling to a continuous processing mode. The lipolytic capability of the enzyme membrane reactor was investigated in fouling tests with olive oil. While the untreated PVDF-Ref samples show a decrease in permeance to 2% of the initial value, the PVDF-*g*-EL demonstrates reduced fouling reaching 32% of initial permeance. The surface chemistry of the enzyme-loaded membrane is significantly changed according to XPS, FTIR spectroscopy, as well as zeta potential, and water contact angle measurements. Subsequently, the self-cleaning properties were studied by activation of the enzymes via impregnating the fouled membranes in aqueous PBS buffer (pH 8, 37 °C). Within 3 h, the lipolytic membranes can regenerate 100% of the filtration performance, in contrast to only 5% of the reference. Reusability was demonstrated by performing three consecutive fouling and self-cleaning cycles, resulting in 95% performance regeneration after the third cycle with a total feed volume of 6550 L m^−2^. Thus, the presented method might be suitable as a promising immobilization strategy for biomacromolecules such as enzymes, enabling the production of biocatalytically active membrane filters to be applied in environmentally-friendly processes.

## Figures and Tables

**Figure 2 membranes-12-00599-f002:**
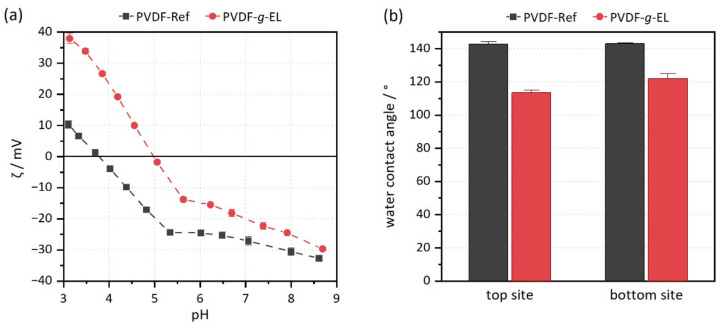
Characterization of surface properties. (**a**) Zeta potential curves; (**b**) contact angles of the top and bottom site of the membrane samples.

**Figure 3 membranes-12-00599-f003:**
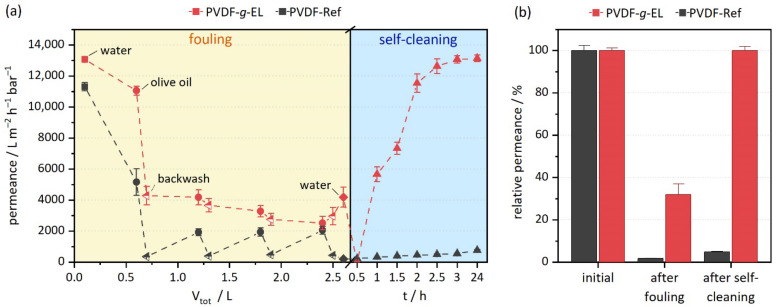
First fouling and self-cleaning cycle for PVDF-Ref and lipolytic PVDF-*g*-EL. (**a**) Permeance curves as a function of total filtered volume, *V_tot_*, during the fouling cycle, and as a function of time, *t*, during the self-cleaning cycle; (**b**) relative permeance *P/P_0_* at the beginning, after fouling, and after self-cleaning.

**Figure 4 membranes-12-00599-f004:**
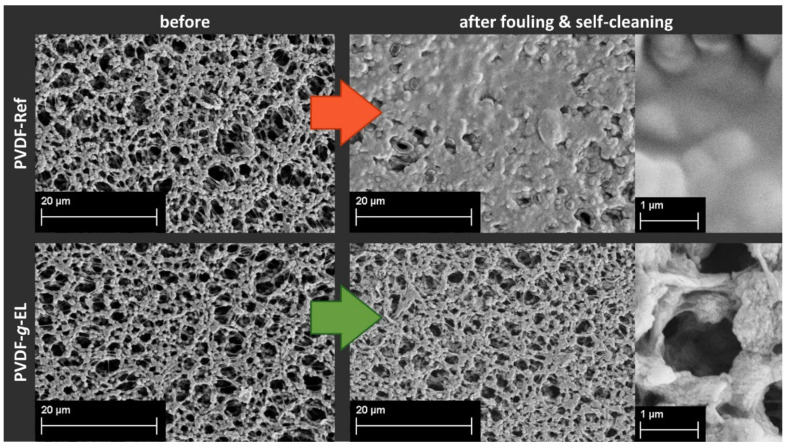
SEM images of pristine PVDF-Ref and modified PVDF-*g*-EL before and after the first fouling and self-cleaning cycle.

**Figure 5 membranes-12-00599-f005:**
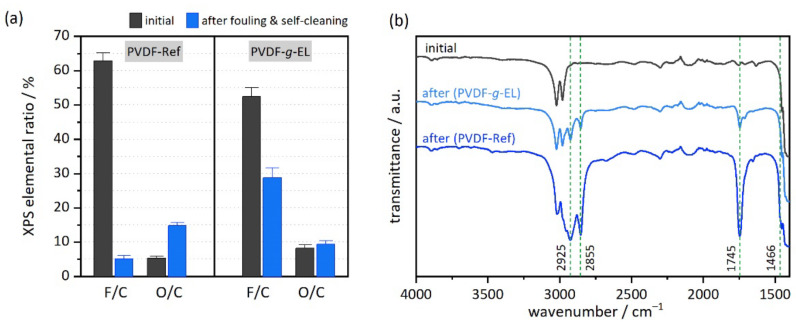
Characterization of samples before and after the first fouling and self-cleaning cycle by using (**a**) XPS; and (**b**) FTIR spectroscopy.

**Figure 6 membranes-12-00599-f006:**
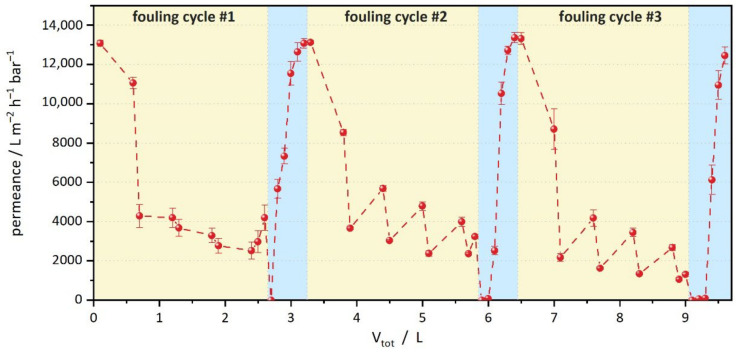
Reusability of lipolytic PVDF-*g*-EL within 3 consecutive fouling and self-cleaning cycles. Fouling cycles are highlighted with a yellow background, self-cleaning cycles (each 3 h) are highlighted with a blue background. PVDF-Ref was not used due to severe fouling after the first cycle.

**Table 1 membranes-12-00599-t001:** Investigated factors of the I-optimal RSM design.

Factor	Name	Units	Type	Lower Limit	Upper Limit
A	enzyme concentration	g L^−1^	numeric	1.0	10.0
B	impregnation time	min	numeric	0.1	10.0
C	irradiation dose	kGy	numeric	50	200
D	enzyme	–	categoric	FE-01	EL-01

**Table 2 membranes-12-00599-t002:** ANOVA results of I-optimal design for *V_max_* (transformed with *ln*).

Source	Sum of Squares	Df	Mean Square	F-Value	*p*-Value
Block	0.067	1	0.067	-	-
Model	76.208	17	4.483	41.24	<0.0001
A: enzyme conc.	16.805	1	16.805	154.59	<0.0001
B: impreg. time	0.106	1	0.106	0.98	0.3314
C: irrad. dose	8.632	1	8.632	79.41	<0.0001
D: enzyme	0.172	1	0.172	1.58	0.2184
AB	0.028	1	0.028	0.26	0.6130
AC	4.305	1	4.305	39.60	<0.0001
AD	0.500	1	0.500	4.60	0.0405
BC	0.053	1	0.053	0.49	0.4891
CD	0.897	1	0.897	8.25	0.0075
A²	2.289	1	2.289	21.06	<0.0001
B²	1.667	1	1.667	15.33	0.0005
C²	3.678	1	3.678	33.84	<0.0001
A²B	1.675	1	1.675	15.40	0.0005
AB²	1.096	1	1.096	10.08	0.0035
B²C	0.683	1	0.683	6.28	0.0181
B³	1.190	1	1.190	10.95	0.0025
C³	0.766	1	0.766	7.04	0.0128
Residual	3.152	29	0.109	-	-
Lack-of-Fit (LOF)	2.735	21	0.130	2.50	0.0922
Pure Error	0.417	8	0.052	-	-
Cor Total	79.428	47	-	-	-

**Table 3 membranes-12-00599-t003:** Kinetic parameters of optimized PVDF-*g*-EL (*n* = 5).

*V_max_*/µM s^−1^	*K_m_*/µM	*V/K*/10^−3^ s^−1^	*A_sp_*/U m^−2^
2.7 ± 0.4	62.9 ± 19.3	44.1 ± 7.7	823 ± 118

**Table 4 membranes-12-00599-t004:** XPS measurement data of pristine PVDF-Ref and enzyme-loaded PVDF-*g*-EL (*n* = 4).

Sample	Elemental Composition/at %						Elemental Ratio/%		
	C	F	O	N	S	Si	F/C	O/C	N/C
PVDF-Ref	59.18	37.21	3.13	0.00	0.00	0.49	62.89	5.29	0.00
	±0.57	±1.05	±0.40	±0.00	±0.00	±0.10	±2.38	±0.62	±0.00
PVDF-*g*-EL	61.51	32.27	5.03	0.58	0.00	0.62	52.47	8.17	0.93
	±0.47	±1.32	±0.67	±0.10	±0.00	±0.36	±2.50	±1.04	±0.15

## Data Availability

Not applicable.

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
