# Peer review of "Reagent-Free Immobilization of Industrial Lipases to Develop Lipolytic Membranes with Self-Cleaning Surfaces"

_membranes, 2022, doi:10.3390/membranes12060599_

Round 1

Reviewer 1 Report

The authors have reported development of biocatalytic membrane reactors, based on lipase enzymes coupled onto a polyvinylidene fluoride. The physico-chemical characterization is based on determination of contact angles and zeta potential. Owing to potential industrial application of such systems to biocatalysis, the contribution would be of interest in the readers of Membranes. 

Author Response

Many thanks for the positive review!

Reviewer 2 Report

membranes-1742276

Reagent-Free Immobilization of Industrial Lipases to Develop  Lipolytic Membranes with Self-Cleaning Surfaces

Martin Schmidt, Andrea Prager, Nadja Schönherr, Roger Gläser and Agnes Schulze

The authors immobilize enzymes onto polymeric membranes without the use of solvent. The method was based on electron beam irradiation. Two industrial lipase enzymes were coupled onto PVDF flat sheet membrane. The membrane was tested and characterized and they found, using olive oil, 100% regeneration of filtration performance after 3 h of self-cleaning in an aqueous buffer and regeneration of 95% with three consecutive cycles

The work is interesting and well designed. It can be accepted for publication after revision.

The novelty of the work should be added in the introduction part. The differences with previous articles from the group should be addressed.

Maybe I am wrong but the optimization of the supported enzyme content should be better presented. I did not understand what the optimum content for supported enzyme is, with respect to fouling time, filtration performance and regeneration.

Generally iep is the pH where the zeta potential at the share plane is zero. The surface charge is controlled from the point of zero charge, pzc. When there is adsorption these two values are different. Please check the lines 326-331.

Line 381 please add the temperature

Author Response

The authors immobilize enzymes onto polymeric membranes without the use of solvent. The method was based on electron beam irradiation. Two industrial lipase enzymes were coupled onto PVDF flat sheet membrane. The membrane was tested and characterized and they found, using olive oil, 100% regeneration of filtration performance after 3 h of self-cleaning in an aqueous buffer and regeneration of 95% with three consecutive cycles

The work is interesting and well designed. It can be accepted for publication after revision.

Our answer: Thank you very much for your time and constructive remarks! The individual points are addressed below.

The novelty of the work should be added in the introduction part. The differences with previous articles from the group should be addressed.

Our answer: In previous work, we have prepared self-cleaning membranes with immobilized enzymes by chemical reactions, as mentioned in lines 347-349. In this study, we attempted to replace the chemical reaction step with an electron beam irradiation step to significantly improve the modification approach. In addition, to the best of our knowledge, it was the first time that a radiation immobilized enzyme membrane reactor was used in self-regenerating experiments.

Maybe I am wrong but the optimization of the supported enzyme content should be better presented. I did not understand what the optimum content for supported enzyme is, with respect to fouling time, filtration performance and regeneration.

Our answer: Thank you for this insightful note. For details on the optimization, please refer to our answer to the editor's question. Briefly, the optimization was performed only for the preparation of the enzyme-modified membranes. It was not performed for the fouling experiments because the time required would be very high! The authors assumed that excellent self-cleaning capability can be achieved if the immobilized enzyme activity is maximized within the experimental limits. Therefore, a Design-of-Experiments approach was used to prepare such enzyme-loaded membranes. By applying the desirability function as a statistical tool, it is possible to maximize the reaction rate (i.e., enzyme activity) of the immobilized enzyme while minimizing the experimental parameters (enzyme con­centration, impregnation time, irradiation dose). Some explanations have been included in the man­uscript (lines 142 and 210).

Generally iep is the pH where the zeta potential at the share plane is zero. The surface charge is controlled from the point of zero charge, pzc. When there is adsorption these two values are different. Please check the lines 326-331.

Line 381 please add the temperature

Our answer: Many thanks for this interesting comment! From our understanding, for the reference membrane, the potential at the surface results from the adsorption of OH anions. According to [Environ. Sci. Technol. 1998, 32, 19, 2815-2819], IEP and PZC are identical in this case since no other ions are involved. Furthermore, the manufacturer of the measuring instruments, Anton Paar, also uses the term IEP. The temperature (37 °C) was added.

Reviewer 3 Report                                                         

The current work focuses on the "Reagent-Free Immobilization of Industrial Lipases to Develop 2 Lipolytic Membranes with Self-Cleaning Surfaces". The experimental work appears to have been carried out well. However, a few points deserve attention for further publication. I suggest that it is accepted for publication after the following revisions:

-ABSTRACT: Please edit the abstract by emphasizing to the very important results of the study.   What parameters were optimized? Authors must include numbers with the results found. How much polymer was utilized to process? Furthermore, what are the conditions of reactions? Temperature, pH, ionic strength, for example. This information should be included in the abstract.

-Too many keywords are used. Some of them are not so important. Please edit this section.

-The introduction section length is very short and this process needs to be explained in more details by reviewing the other researchers work the introduction of the manuscript.

-Please provide the schematic illustration for the proposed process. (including the PVDF and which bonds break)

-Please provide the mechanism for illustrated scheme. -In the structure of PVDF two kinds of bonds are present. By comparison of these bonds, very one know that the C-H bond is weaker than C-F bond.

-Please provide the XPS data for pristine PVDF and insert in table 4. Also please compare the elemental analysis results with each other's. provide comparison is not meaningful. 

-The elemental analysis technique is every sensitive to molecular impurity, please describe, how the author confirms the purity of the samples?

-I suggest that the author used the Leucine amino acid as simple molecule before use the reference BSA sample.

-Please do the solvent-non solvent technique to purify the polymer after immobilization of enzyme.

-Please provide the H-NMR and F-NMR of the purified grafted sample, and compare with virgin PVDF sample.

-Please describe why the content of "N" in the PVDF-BSA (PVDF-Ref) is equal to zero.

-The results of XPS for nitrogen atom for sample PVDF-g-EL showed that the very low degree of substitution was occurred in the PVDF structure based on 0.58% for N. How the author describe the reasons of low graft yield? And based on these results, please describe the advantages of this grafting technique and compare with chemical method.   

Author Response

The current work focuses on the "Reagent-Free Immobilization of Industrial Lipases to Develop 2 Lipolytic Membranes with Self-Cleaning Surfaces". The experimental work appears to have been carried out well. However, a few points deserve attention for further publication. I suggest that it is accepted for publication after the following revisions:

Our answer: Thank you very much for your time and comprehensive remarks. The specific comments will be addressed individually.

ABSTRACT: Please edit the abstract by emphasizing to the very important results of the study. What parameters were optimized? Authors must include numbers with the results found. How much polymer was utilized to process? Furthermore, what are the conditions of reactions? Temperature, pH, ionic strength, for example. This information should be included in the abstract.

Our answer: Some additional information is now included in the abstract.

Too many keywords are used. Some of them are not so important. Please edit this section.

Our answer: Some keywords were removed.

The introduction section length is very short and this process needs to be explained in more details by reviewing the other researchers work the introduction of the manuscript.

Our answer: Many thanks for this helpful comment. The introduction part has been revised to clarify some aspects (lines 49-54, 62-67). Please note that further details of the electron beam-based process have been covered in other parts of the manuscript for better readability, e.g., the general procedure was discussed in the experimental part (section 2.2), and the mechanism was stated in the results part (lines 265-281).

Please provide the schematic illustration for the proposed process. (including the PVDF and which bonds break). Please provide the mechanism for illustrated scheme.

Our answer: Schemes and mechanisms are indeed important information to understand the process. The authors would like to point out that previous studies have been published that contain all the necessary information. To avoid overlap with other publications, it was decided not to republish some images. Instead, references were added in the manuscript to provide the information, e.g., about the processing (line 100; Front Chem 2021, 9, 804698, doi:10.3389/fchem.2021.804698), or about the mechanism (lines 265-281: Polymers 2021, 13, 1849, doi:10.3390/polym13111849, and Front Chem 2021, 9, 804698, doi:10.3389/fchem.2021.804698).

In the structure of PVDF two kinds of bonds are present. By comparison of these bonds, very one know that the C-H bond is weaker than C-F bond. Please provide the XPS data for pristine PVDF and insert in table 4. Also please compare the elemental analysis results with each other's. provide comparison is not meaningful. 

Our answer: The XPS data of pristine (untreated, virgin) PVDF are already included in Table 4! Please note that the sample labeled as "PVDF-Ref" is actually the untreated PVDF membrane used as the base material within this study. In addition, the “elemental composition” reported in Table 4 has already been compared in the manuscript text (lines 323-333). The increase or decrease of the values was mentioned therein and explained according to other studies: Journal of Membrane Science 2018, 563, 481-491, doi:10.1016/j.memsci.2018.06.013; Reactive and Functional Polymers 2013, 73, 698-702, doi:10.1016/j.reactfunctpolym.2013.02.013; Polymers 2015, 7, 1837-1849, doi:10.3390/ polym7091485. The “elemental ratio” given in Table 4 is a different way of representing the XPS data and is provided as an additional service to the readers of the manuscript.

The elemental analysis technique is every sensitive to molecular impurity, please describe, how the author confirms the purity of the samples?

Our answer: The standard chemicals used were purchased at a high level of purity, and ultra-pure water from a Millipore® unit was used. The membrane itself was a commercially available product, which although of good quality, is still a technical product. To confirm purity or homogeneity, mult­iple spots (4-5 per sample) were analyzed with XPS and averaged. In this way, the effects of impurities are considered, but they are negligible based on the data. For the enzymes, which were also technical products, only one batch was used to avoid different compositions.

I suggest that the author used the Leucine amino acid as simple molecule before use the reference BSA sample.

Our answer: The reviewer's suggestion is gratefully acknowledged and will be considered in future work. In a previous publication, the amino acid glycine (as well as taurine) was grafted onto PVDF membranes (Polymers 2021, 13, 1849, doi:10.3390/polym13111849). In another recent work, the protein BSA was coupled to a PVDF membrane (Front Chem 2021, 9, 804698, doi:10.3389/ fchem.2021.804698). However, neither single amino acids nor BSA proteins were used as grafting compounds within this study.

Please do the solvent-non solvent technique to purify the polymer after immobilization of enzyme.

Our answer: We will consider this technique in future studies, but it should be noted that enzymes are well-known to be very sensitive to solvents. The use of different solvents could significantly alter the structure and thus the activity of the enzymes, which is beyond the scope of this study.

Please provide the H-NMR and F-NMR of the purified grafted sample, and compare with virgin PVDF sample.

Our answer: NMR spectroscopy has been extensively discussed and explored at our institute in previous work. However, it was not possible to obtain useful data because the use of polymers poses several issues: first, the PVDF membrane is poorly soluble and requires special methods, and second, even when we finally prepared it, the bulk signals were very pronounced compared to the surface signals. However, the latter are especially important since we perform surface grafting reactions. Therefore, NMR studies are far beyond the scope of this study.

Please describe why the content of "N" in the PVDF-BSA (PVDF-Ref) is equal to zero.

Our answer: Thank you for this question. It seems that there is a misunderstanding involved. The sample noted as "PVDF-Ref" is indeed only the pristine (untreated, virgin) PVDF membrane. Accordingly, the N content must be 0, which also proves the purity of the material. No PVDF-BSA membrane was prepared in this study. This could be confused with one of our previous studies: Front Chem 2021, 9, 804698, doi:10.3389/fchem.2021.804698.

The results of XPS for nitrogen atom for sample PVDF-g-EL showed that the very low degree of substitution was occurred in the PVDF structure based on 0.58% for N. How the author describe the reasons of low graft yield? And based on these results, please describe the advantages of this grafting technique and compare with chemical method.

Our answer: Despite the low N content, which was attempted to be explained in the manuscript in sections 3.1 and 3.2, the enzyme activity was very high. The reasons for the reported values are part of further work, but it appears that due to the buffer ingredients of the industrial enzymes, fewer cross-linking and binding events may have occurred essentially. This reduces the grafting yield to some extent. The advantages and disadvantages of this method compared with chemical methods were briefly mentioned in the introduction (lines 66-70), but also discussed in detail in some of the references given. In short, the radiation method is much faster, cheaper, cleaner, and in some cases can result in higher grafting yields (Front Chem 2021, 9, 804698, doi:10.3389/fchem.2021.804698). Overall, this method is quite new and there is not so much data available yet. In contrast, chemical-based methods seem to achieve good grafting yields more easily for different types of enzymes. This is still part of active research.

Round 2

Reviewer 3 Report

-